# Peptide-oligourea hybrids analogue of GLP-1 with improved action in vivo

Juliette Fremaux[1], Claire Venin[1], Laura Mauran[1], Robert H. Zimmer[1], Gilles Guichard [ID] [2] &
Sébastien R. Goudreau [ID] [1]

Peptides have gained so much attention in the last decade that they are now part of the main strategies, with small molecules and biologics, for developing new medicines. Despite substantial progress, the successful development of peptides as drugs still requires a number of limitations to be addressed, including short in vivo half-lives and poor membrane permeability. Here, we describe the use of oligourea foldamers as tool to improve the pharmaceutical properties of GLP-1, a 31 amino acid peptide hormone involved in metabolism and glycemic control. Our strategy consists in replacing four consecutive amino acids of GLP-1 by three consecutive ureido residues by capitalizing on the structural resemblance of oligourea and α-peptide helices. The efficacy of the approach is demonstrated with three GLP-1-oligourea hybrids showing prolonged activity in vivo. Our findings should enable the use of oligoureas in other peptides to improve their pharmaceutical properties and may provide new therapeutic applications.

[1] UREKA—ImmuPharma Group, 2 rue Robert Escarpit, 33607 Pessac, France. [2] Univ. Bordeaux, CNRS, CBMN, UMR 5248, Institut Européen de Chimie et Biologie, 2 rue Robert Escarpit, 33607 Pessac, France. Correspondence and requests for materials should be addressed to G.G.(email: g.guichard@iecb.u-bordeaux.fr) or to S.R.G.(email: sebastien.goudreau@immupharma.com)

During the past decade, peptide therapeutics have gained considerable attention in pharmaceutical research and development (R&D)[1–4]. Indeed, peptides have proved to be valuable tools to access extra-cellular targets with medium to large active sites and they are now intensively investigated to access intracellular protein–protein interaction (PPI) targets, a very important topic in recent pharmaceutical research[5–10]. This is remarkable considering peptides have important shortcomings as they generally show poor membrane permeability, poor bioavailability, and short in vivo half-life.

Much efforts have been invested in peptide and peptidomimetic chemistries[6–11] to address those weaknesses in the hope of finding an alternative to peptides and recently foldamer research has attracted much interest[12–15]. Oligoureas are in the limited list of such potential foldamers as they offer 3-D space similarity, metabolic compatibility, water solubility, and flexibility of functionalities[16–21]. Like peptides, they are easily synthesised by iterative coupling on solid support and possess their own secondary, tertiary and quaternary structures based on their sequences[22,23]. Most importantly, the oligourea backbone is resistant to proteases[20] and can be interfaced with peptide α-helices as it adopts an helical conformation that does not disrupt the peptide α-helix propagation[16,21,24]. This is noteworthy as (1) many biologically active peptides contain an α-helix (a large fraction of PPIs involve an α-helix) and (2) those helix portions could potentially be replaced or partially replaced by oligoureas with minimal effect on the binding while improving the proteolytic resistance of the peptide. Such strategy would be a valuable tool to design peptide therapeutics as their pharmaceutical properties could be improved. Further investigation was however needed, as the compatibility of peptide-oligourea hybrids in biological systems and their utilisation in vivo were undocumented.

GLP-1 (glucagon-like peptide-1) is an endocrinal peptide hormone expressed by the pancreas and by many other organs including the gastrointestinal tract, the heart and the brain[25,26]. It is released primarily in response to food intake and controls the secretion of insulin in function of the glucose level. Its receptor, GLP-1R is a member of class B GPCRs (G protein-coupled receptors) which are characterized by a seven transmembrane domain at their C-termini and a large extra-cellular domain at their N-termini[27–30]. GLP-1, binds to GLP-1R mostly in an α-helix conformation with a large surface of contact with both the extra-cellular domain and the seven transmembrane domain[27,30]. In vivo, GLP-1 has a half-life of only 2–3 min, preventing its therapeutic utilisation[31–35]. The identified proteases responsible for its rapid decay are DPP-4 (dipeptidyl peptidase-4)[34], which cleaves selectively between residues 2 and 3, and NEP 24.11[35], which cleaves at multiple sites. Different strategies were used to improve the peptide half-life including sequence modifications (exenatide, lixisenatide)[36,37], conjugation to molecules with propensity to bind albumin (liraglutide, semaglutide)[36,37], fusion to large proteins (albiglutide, dulaglutide, efpeglenatide)[36,37], side chain cross linking[38,39], and α → β-residue replacement[40–42] (Fig. 1a).

Owing to its α-helical binding conformation, GLP-1 stroke us as both a good model for testing α-helix mimicry with oligourea foldamers and a relevant pharmaceutical target. Indeed GLP-1R agonists proved to be efficient for the treatment of type 2 diabetes mellitus[37] and studies indicate they could be used in other indications[26,43,44] such as obesity[45,46], adverse cardiovascular events[25], cognitive disorders[44] and non-alcoholic steatohepatitis (NASH)[47].

Herein we report a series of GLP-1-oligourea hybrids with moderate to excellent affinity to GLP-1R including three hybrids with prolonged action in mice.

## Results and discussion

**Peptide-oligourea hybrids design and functional assay.** Our peptide–oligourea hybrids design started with GLP-1-NH$_2$ and the introduction of a glycine in position 2 of GLP-1 instead of alanine (GLP-1-G$^2$, **1**) as it is known to prevent DPP-4 degradation[31–33]. We then identified the amount of consecutive amino acid residues that can be replaced by ureido units. This aspect was important considering that GLP-1 has key interactions with GLP-1R at both ends of the peptide[27,30]. If an oligourea portion placed in the middle of the peptide induced a torsion or an elongation that is not exactly the same as the polypeptide it replaces, a drastic negative impact on the affinity would be expected. We thus decided to start our investigation by replacing four consecutive amino acids (X$_4$) with three consecutive ureido residues (X$^u_3$) based on our model, suggesting that this combination gives the best overlaps of the terminal atoms (Fig. 1b). It is noteworthy that the diameter of the triad helix turn is slightly larger than the peptide α-helix turn and it should be taken into account when designing hybrids.

Next, we turned our attention to the side chains of the removed tetrapeptide. Our model suggested that the projection of the side chains of X$^u$ residues would not be the same as the native X residues they replaced (Fig. 1c). Indeed, examination of the model suggested that the side chain of X$^u$1 did not project in the same direction as in X1. However, it suggested that a ureido residue with its side chain shifted to the second backbone methylene (i.e. α-C)[18] would give a similar projection. The second side chain (X$^u$2) gave the best fit with the corresponding α-amino acid side chain (X2) based on this model. On the other hand, the third amino acid side chain seemed to be the least imitable as it is superimposed with a urea nitrogen of the third ureido residue X$^u$3. However, our model also suggested that this third ureido residue could mimic the fourth amino acid residue (X4) by changing the substitution pattern (shift to the second methylene —α-C). Overall, our model suggested that mimicking the precise projection of α-amino acid side chains was possible although not necessary trivial.

Considering the relative difficulty to select the most appropriate side chains in ureido units to mimic a given tetrapeptide segment with high fidelity, we thus decided to start with an Ala$^u$ triad (A$^u$A$^u$A$^u$) scan for simplification. The objective was to avoid any negative impact of the amino urea side chains on the potency so the observed activity would correlate the loss of affinity predicted from the Ala scan (Table 1 and Fig. 2)[48,49]. A lower activity than predicted would mean a negative impact coming from the backbone modification. In our first round of syntheses, the known helical portion of GLP-1 spanning residues 9–31 was scanned with Ala$^u$ triads (A$^u$A$^u$A$^u$). We avoided, however, to synthesise hybrids lacking D9 or F22, considering that these amino acids are requisite for potency (Table 1). All compounds were synthesized using standard solid-phase synthesis techniques and microwave assistance[22].

The agonist activity of these hybrids were obtained by functional assays using cells expressing the GLP-1R and by measuring the receptor-mediated cAMP produced. As expected[31,32], GLP-1-G$^2$ (**1**) was 2.5 times less potent than GLP-1, with still a 50% effective concentration (EC$_{50}$) of 0.24 nM. Interestingly, many hybrids from the Ala$^u$ triad scan proved to be potent, demonstrating the good overlaps of the oligourea backbone with the peptide backbone. The most active hybrids were selected and native side chains were introduced in the oligourea triads in new rounds of synthesis with focus on recovering the most important side chain interactions: D9, Y13, E15, F22 and I23. Table 1 shows a selection of the most representative results obtained in this study. Interestingly, in most cases the reintroduction of the native side chains improved the

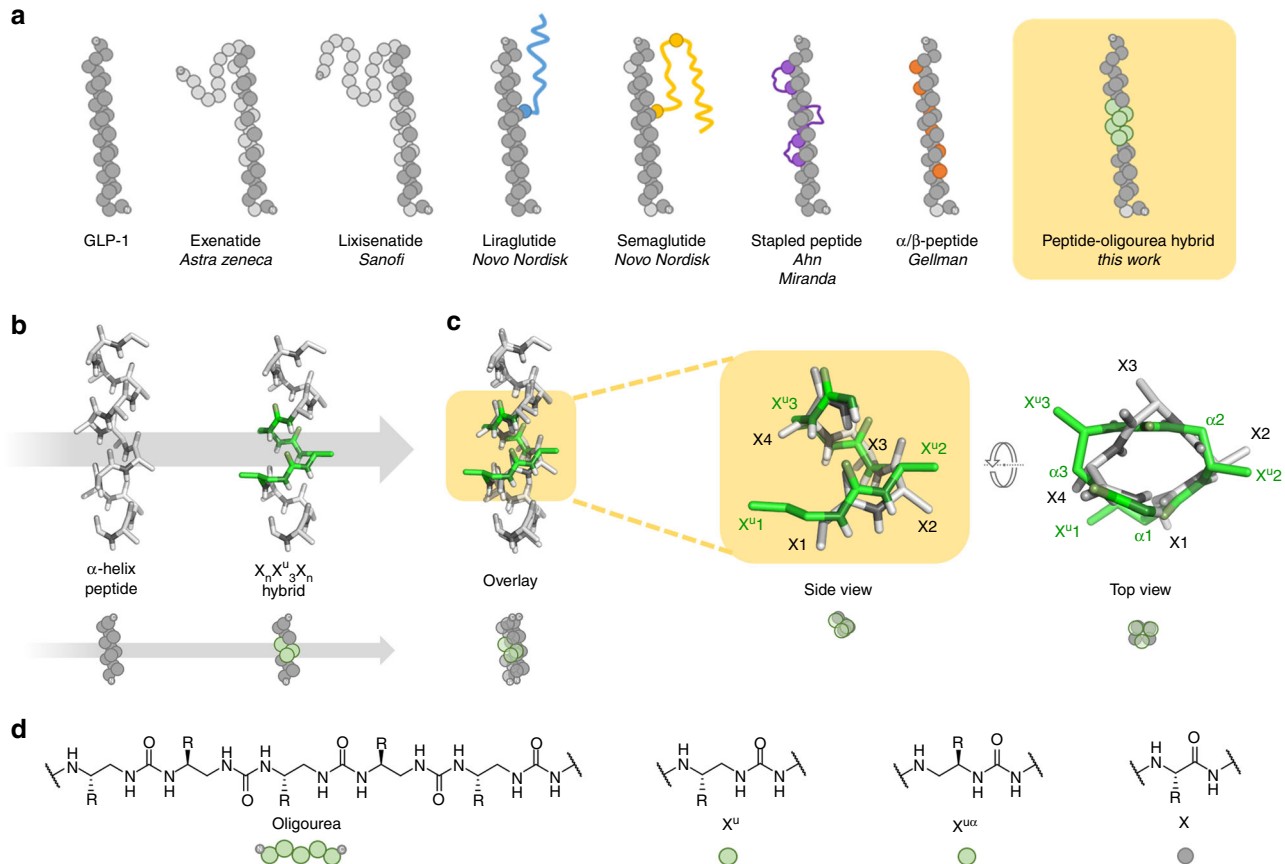

**Fig. 1** Structural analysis. **a** Schematic representation of different GLP-1 analogues previously reported and of the present approach based on peptide-oligourea hybrids. **b–c** Comparison of α-helical and oligourea backbones. **b** Overlay (side-views) of a peptide oligourea hybrid (derived from the crystal structures of helical peptide-oligourea hybrids[16]) over a peptide. **c** Overlay (side- and top-views) of an oligourea triad (extracted from (**b**), green backbone) over a tetrapeptide in α-helical conformation (grey backbone). **d** Chemical structures of an oligourea backbone, as well of $X^u$, $X^{u\alpha}$, and X residues used in this study and their associated cartoon representations

potency of the GLP-1-oligourea hybrids. GG2[$Y^uE^uA^u$][14–17] (**9**) even gave a better affinity (0.18 nM) than the native peptide **1** (0.24 nM). It is noteworthy that hybrids GG2[$D^{u\alpha}A^uA^u$][9–12] (**2**, 13 nM) and GG2[$F^{u\alpha}I^uA^u$][22–25] (**16**, 62 nM) which contain native side chains and shifted substitution pattern still preserve substantial affinity with GLP-1R, although with over a hundred fold loss compared to the native peptide. Nevertheless, considering the importance of the modifications, the mere observation that oligourea triads can mimic those capital interactions is remarkable. We also investigated the possibility to introduce two successive triads by synthesizing compound GG2[$Y^uE^uA^uA^uA^uA^u$][14–21] (**23**), which comes from the combination of **9** (0.18 nM) and **14** (4.1 nM). We were delighted to find that **23** was still active (3.4 nM) with a similar potency to **14**, considering that 25% of the GLP-1 sequence was replaced by oligoureas.

**Enzymatic and mouse plasma degradation studies.** Having in hands active hybrids, we then turn our attention to their stability toward proteolysis. Since homooligoureas are fully resistant to proteases[20], improved proteolytic protection was expected for hybrids compare to their native peptide **1**. As mentioned above, the two main identified proteases for the degradation of GLP-1 in vivo are DPP-4 and NEP-24.11. As the glycine in position 2 of our analogues prevents the specific DDP-4 cleavage between residues 2 and 3, our attention was turned toward the later. NEP 24.11 (neutral endopeptidase 24.11, also known as Neprilysin, CD10, MME and CALLA) is an unspecific membrane protease

that quickly cleaves GLP-1 at multiple sites[35]. It was showed that a GLP-1 analogue with improved half-life in a NEP 24.11 degradation assay had prolonged action in mice, therefore making this assay relevant to evaluate the potential efficacy in vivo of our compounds[40]. In addition to NEP 24.11, mouse plasma was also utilized to assess the stability of our compounds.

The half-lives of compounds **1**, **2**, **5**, **9**, **11**, **14**, **16**, **22** and **23** in presence of NEP 24.11 and in presence of mouse plasma were determined by following the decay of the analogues by LCMS (Table 2). In the NEP 24.11 degradation assay, the native peptide **1** had a half-life of 3.6 h and unfortunately the hybrids **5**, **9** and **16** showed diminished half-lives of 2.9, 3.0 and 1.8 h, respectively. Although these $t_{1/2}$ values were not significantly different from the $t_{1/2}$ of peptide **1** (3.6 h), statistical significance was obtained at the 4 h time point for these three hybrids demonstrating their lower stability (Supplementary Fig. 1). Those results were intriguing, as we did not expect lower proteolytic protection for some hybrids. Further investigations showed no indication that the cleavage sites for the destabilized hybrids were located in or next to the oligourea triads making it more puzzling. Nevertheless, in two cases (**22**, **23**) the permutation of four amino acids for an oligourea triad improved significantly the proteolytic protection of the hybrids (Table 2 and Supplementary Fig. 1). It is also noteworthy that the analogue **23** where eight amino acids were replaced by two triads had a significant increase of its half-life.

In mouse plasma degradation assay, the native peptide **1** had a half-life of 16 min and hybrids **2**, **5** and **9** had similar $t_{1/2}$ of 17, 23

**Table 1 Bioactivity of GLP-1 and cognate peptide–oligourea analogues in cAMP production functional assay**

| Sequence | Compound | EC$_{50}$ (nM)[a] | SE (nM)[b] | Potency (%)[c] |
|---|---|---|---|---|
| GLP-1  H H A E G T F T S D V S S Y L E G Q A A K E F I A W L V K G R G OH | GLP-1 | 0.10[d] | 0.01[e] | 250 |
| **1**  H H G E G T F T S D V S S Y L E G Q A A K E F I A W L V K G R G NH₂ | GLP-1-G² (GG2) | 0.24 | 0.06 | 100 |
| **2**  H H G E G T F T S DᵘᵃAᵘAᵘ Y L E G Q A A K E F I A W L V K G R G NH₂ | GG2[DᵘᵃAᵘAᵘ]⁹⁻¹² | 13 | 3 | 2 |
| **3**  H H G E G T F T S D AᵘAᵘAᵘ L E G Q A A K E F I A W L V K G R G NH₂ | GG2[AᵘAᵘAᵘ]¹⁰⁻¹³ | 650 | 100 | 0.04 |
| **4**  H H G E G T F T S D V AᵘAᵘAᵘ E G Q A A K E F I A W L V K G R G NH₂ | GG2[AᵘAᵘAᵘ]¹¹⁻¹⁴ | 97 | 9 | 0.2 |
| **5**  H H G E G T F T S D V AᵘYᵘAᵘ E G Q A A K E F I A W L V K G R G NH₂ | GG2[AᵘYᵘAᵘ]¹¹⁻¹⁴ | 1.6 | 0.5 | 15 |
| **6**  H H G E G T F T S D V S AᵘAᵘAᵘ G Q A A K E F I A W L V K G R G NH₂ | GG2[AᵘAᵘAᵘ]¹²⁻¹⁵ | >10000 | N/A | < 0.01 |
| **7**  H H G E G T F T S D V S S AᵘAᵘAᵘ Q A A K E F I A W L V K G R G NH₂ | GG2[AᵘAᵘAᵘ]¹³⁻¹⁶ | >10000 | N/A | < 0.01 |
| **8**  H H G E G T F T S D V S S Y AᵘAᵘAᵘ A A K E F I A W L V K G R G NH₂ | GG2[AᵘAᵘAᵘ]¹⁴⁻¹⁷ | 353 | 58 | 0.07 |
| **9**  H H G E G T F T S D V S S Y YᵘEᵘAᵘ A A K E F I A W L V K G R G NH₂ | GG2[YᵘEᵘAᵘ]¹⁴⁻¹⁷ | 0.18 | 0.07 | 135 |
| **10**  H H G E G T F T S D V S S Y L AᵘAᵘAᵘ A K E F I A W L V K G R G NH₂ | GG2[AᵘAᵘAᵘ]¹⁵⁻¹⁸ | 123 | 23 | 0.2 |
| **11**  H H G E G T F T S D V S S Y L EᵘAᵘAᵘ A K E F I A W L V K G R G NH₂ | GG2[EᵘAᵘAᵘ]¹⁵⁻¹⁸ | 1.9 | 0.4 | 13 |
| **12**  H H G E G T F T S D V S S Y L E AᵘAᵘAᵘ K E F I A W L V K G R G NH₂ | GG2[AᵘAᵘAᵘ]¹⁶⁻¹⁹ | 240 | 36 | 0.1 |
| **13**  H H G E G T F T S D V S S Y L E G AᵘAᵘAᵘ E F I A W L V K G R G NH₂ | GG2[AᵘAᵘAᵘ]¹⁷⁻²⁰ | 1603 | 187 | 0.01 |
| **14**  H H G E G T F T S D V S S Y L E G Q AᵘAᵘAᵘ F I A W L V K G R G NH₂ | GG2[AᵘAᵘAᵘ]¹⁸⁻²¹ | 4.1[f] | 0.9[e] | 6 |
| **15**  H H G E G T F T S D V S S Y L E G Q A AᵘAᵘAᵘ I A W L V K G R G NH₂ | GG2[AᵘAᵘAᵘ]¹⁹⁻²² | >10000 | N/A | < 0.01 |
| **16**  H H G E G T F T S D V S S Y L E G Q A A K E FᵘᵃᵘIᵘAᵘ L V K G R G NH₂ | GG2[FᵘᵃᵘIᵘAᵘ]²²⁻²⁵ | 62 | 28 | 0.4 |
| **17**  H H G E G T F T S D V S S Y L E G Q A A K E F AᵘAᵘAᵘ V K G R G NH₂ | GG2[AᵘAᵘAᵘ]²³⁻²⁶ | 3926 | 2751 | 0.01 |
| **18**  H H G E G T F T S D V S S Y L E G Q A A K E F I AᵘAᵘAᵘ K G R G NH₂ | GG2[AᵘAᵘAᵘ]²⁴⁻²⁷ | >10000 | N/A | < 0.01 |
| **19**  H H G E G T F T S D V S S Y L E G Q A A K E F I A AᵘAᵘAᵘ G R G NH₂ | GG2[AᵘAᵘAᵘ]²⁵⁻²⁸ | 368 | 169 | 0.07 |
| **20**  H H G E G T F T S D V S S Y L E G Q A A K E F I A W AᵘAᵘAᵘ R G NH₂ | GG2[AᵘAᵘAᵘ]²⁶⁻²⁹ | 26 | 6 | 0.9 |
| **21**  H H G E G T F T S D V S S Y L E G Q A A K E F I A W L AᵘAᵘAᵘ G NH₂ | GG2[AᵘAᵘAᵘ]²⁷⁻³⁰ | 2.8 | 0.9 | 8 |
| **22**  H H G E G T F T S D V S S Y L E G Q A A K E F I A W L V AᵘAᵘAᵘ NH₂ | GG2[AᵘAᵘAᵘ]²⁸⁻³¹ | 1.2 | 0.3 | 19 |
| **23**  H H G E G T F T S D V S S Y YᵘEᵘAᵘ AᵘAᵘAᵘ F I A W L V K G R G NH₂ | GG2[YᵘEᵘAᵘAᵘAᵘAᵘ]¹⁴⁻²¹ | 3.4 | 0.8 | 7 |

Oligourea inserts (triads and hexads) are shown as green boxes. The highlighted amino acids in the GLP-1 sequence generated the highest decrease of activity in the Ala scan: red = very high decrease, orange = high decrease. Unless otherwise stated, the data are EC50 values ± SE obtained by non-linear regression on 8 points concentration–response curves performed in duplicates ($n = 2$). Source data are provided as a Source Data file
EC$_{50}$ half maximal effective concentration, cAMP cyclic adenosine monophosphate
[a]GLP-1R potency (EC$_{50}$)
[b]Standard error (SE) on the EC$_{50}$
[c]Percentage of potency compare to GLP-1-G²
[d]Mean value of 26 experiments
[e]Standard error of the mean (SEM) on the EC$_{50}$
[f]Mean value of four experiments.

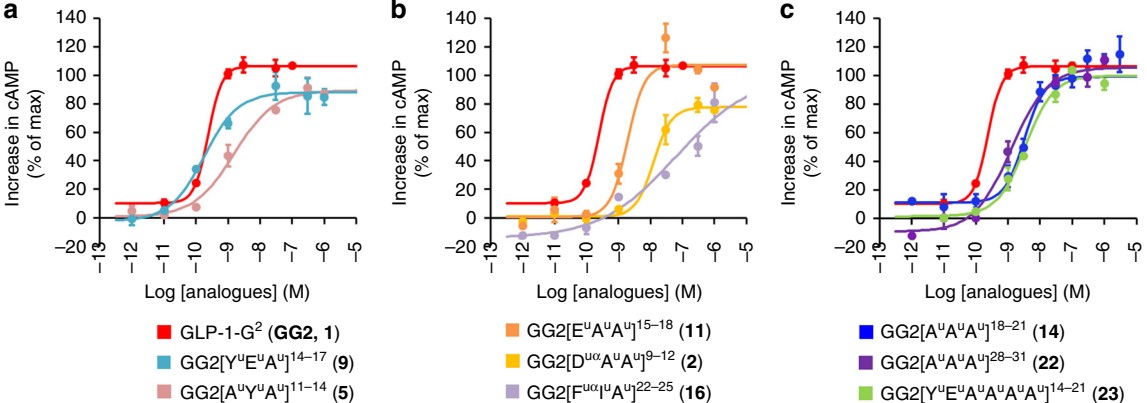

**Fig. 2** Concentration–response curves for selected analogues. Concentration–response curves for selected analogues to induce increase in cAMP in βTC6 cells expressing the human GLP-1R. The data are mean ± SEM of a typical experiment performed twice ($n = 2$). Source data are provided as a Source Data file. cAMP cyclic adenosine monophosphate

**Table 2 NEP 24.11 and mouse plasma degradation assays**

| | Compound | Potency (%)[a] | NEP 24.11 $t_{1/2}$ (h)[b] | SEM (h)[c] | Mouse plasma $t_{1/2}$ (min)[d] | SEM (min)[e] | In vivo activity[f] |
|---|---|---|---|---|---|---|---|
| **1** | GLP-1-G²-NH₂ (GG2) | 100 | 3.6 | 0.1 | 16 | 1 | + |
| **2** | GG2[DᵘᵅAᵘAᵘ]⁹⁻¹² | 2 | 4.4 | 0.4 | 17 | 1 | |
| **5** | GG2[AᵘYᵘAᵘ]¹¹⁻¹⁴ | 15 | 2.9 | 0.3 | 23 | 3 | + |
| **9** | GG2[YᵘEᵘAᵘ]¹⁴⁻¹⁷ | 135 | 3.0 | 0.2 | 17 | 1 | + |
| **11** | GG2[EᵘAᵘAᵘ]¹⁵⁻¹⁸ | 13 | 3.9 | 0.4 | 31* | 3 | |
| **14** | GG2[AᵘAᵘAᵘ]¹⁸⁻²¹ | 6 | 4.9 | 0.8 | 38*** | 2 | +++ |
| **16** | GG2[FᵘᵅIᵘAᵘ]²²⁻²⁵ | 0.4 | 1.8 | 0.1 | 38*** | 5 | |
| **22** | GG2[AᵘAᵘAᵘ]²⁸⁻³¹ | 19 | 4.5 | 0.5 | >60*** | na | +++ |
| **23** | GG2[YᵘEᵘAᵘAᵘAᵘAᵘ]¹⁴⁻²¹ | 7 | 6.1* | 0.9 | 39*** | 5 | +++ |

Statistic by one-way ANOVA with Dunnett's multiple comparison test: *$p < 0.05$; ***$p < 0.001$, comparing peptide **1** to oligomers. Source data are provided as a Source Data file
[a]Percentage of potency (cAMP production) compared to GLP-1-G² (see Fig. 2)
[b]Half-life of GLP-1 analogue in NEP 24.11 degradation assay: mean value of three replicates ($n = 3$)
[c]Standard error of the mean on the NEP 24.11 $t_{1/2}$
[d]Half-life of GLP-1 analogue in mouse plasma degradation assay: mean value of four replicates ($n = 4$)
[e]Standard error of the mean on the mouse plasma $t_{1/2}$
[f]Indicator of in vivo activity: + active, +++ more active

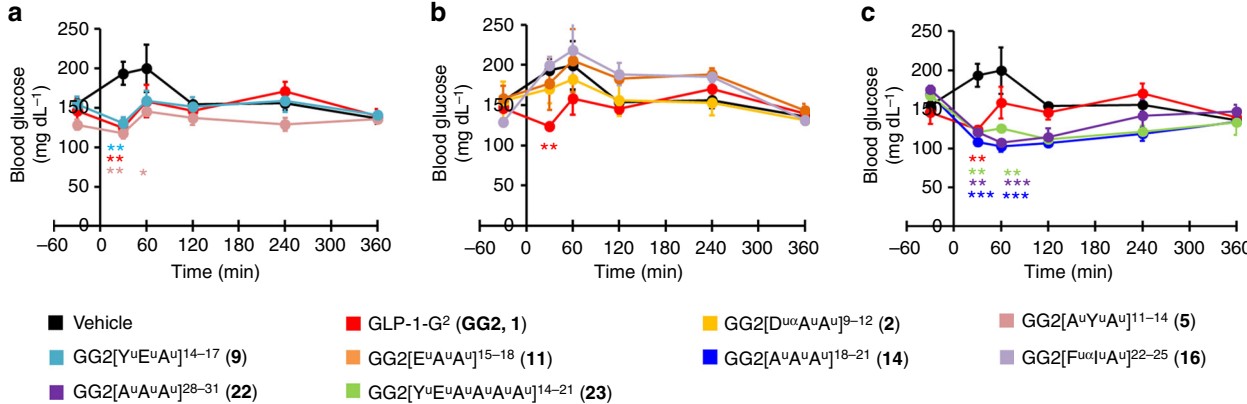

**Fig. 3** Blood glucose study in mice. **a–c** Blood glucose concentration in normal mice (C57BL/6J, male, 20–25 g, $n = 3$) before and after dosing: 5 µg per mouse (50 nmol kg⁻¹) i.v. Formulation: 20 µg mL⁻¹ in PBS 1×. The dosing was done at T0. Data are mean ± SEM. Statistic by two-way ANOVA and Bonferroni post-test: *$p < 0.05$, **$p < 0.01$, ***$p < 0.001$, comparing vehicle to oligomers. Source data are provided as a Source Data file

and 17 min, respectively. Interestingly, hybrids **11**, **14**, **16**, **22** and **23** had significantly longer half-lives. In the case of **22**, a half-life of >60 min was observed, which represents over 3.8-folds improvement compared to peptide **1**. Over the eight hybrids selected, five had improved stability in mouse plasma with the insertion of the triad at various positions on the peptide. The proteolytic stability to NEP 24.11 could predict mostly the results in mouse plasma. The next step was to investigate if these improvements in half-lives could be transposed in vivo.

**In vivo studies**. Although improving the proteolytic protection or the plasmatic stability of some hybrids represented a significant milestone, the most important question remained: are those improvements reflected in vivo? To answer this important question we conducted three sets of studies.

In the first study, we evaluated the effect of the analogues on the blood glucose concentration of normal mice (Fig. 3). The fed mice were treated with the analogues and their blood glucose concentration was monitored before and after dosing for 6 h. The native peptide GLP-1-G² (**1**) showed a decreased of blood glucose concentration after 30 min, proving its efficacy in vivo, although for a short time as a rapid raise to the initial concentration level is observed after 1 h. Hybrids **5** and **9** showed efficacy but no improvement, indicated by a similar blood glucose curve to

peptide **1** (Fig. 3a). Hybrids **2**, **11** and **16** showed no activity (Fig. 3b) since no initial drop in blood glucose concentration was observed and the curves resembled the vehicle one. Most interestingly, hybrids **14**, **22** and **23** showed an improved efficacy compare to the native peptide **1** (Fig. 3c). Indeed, a decrease of blood glucose concentration is observed after 30 min and the effect is preserved for 4 h after dosing.

These results were well predicted by the plasma stability of the hybrids as all the compounds (**2**, **5** and **9**) with similar plasma stability to peptide **1** gave comparable results in vivo. The compounds with lower potencies **2** (2%) and **16** (0.4%) showed no effect after 30 min and were therefore less effective than peptide **1** (100%), although **16** had longer half-life in plasma (16 vs 38 min). This indicates that passing a certain threshold, the potency is just not good enough to produce an observable effect. In the case of hybrid **11**, no effect was observed on the blood glucose after 30 min of the injection. These results were more difficult to explain as the mouse plasma half-life of **11** is prolonged (31 min) and the potency (13%) is comparable to **14** (6%), **22** (19%) and **23** (7%). As expected, the three best compounds in vivo, **14**, **22** and **23**, had all (1) improved stability in NEP 24.11 degradation assays, (2) prolonged half-lives in mouse plasma and (3) decent potency. This is noteworthy as these analogues are at least 10-fold less potent then GLP-1-G² (**1**), meaning that even if at least 10 times more compound is required

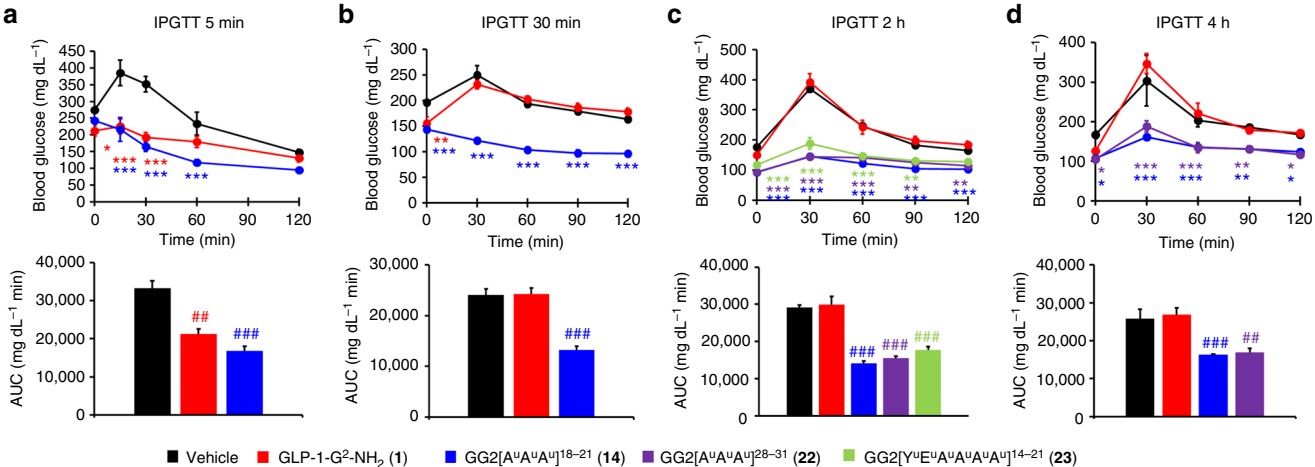

**Fig. 4** IPGTT studies in mice. **a–d** Intraperitoneal glucose tolerance test (IPGTT) at different time after dosing in fasted normal mice (C57BL/6J, male, 20–25 g). Dosage: 200 μg kg$^{-1}$ (50 nmol kg$^{-1}$) i.v. Formulation: 20 μg mL$^{-1}$ in PBS 1×. IPGTT: glucose 2 g kg$^{-1}$ i.p. at T0. **a** IPGTT after 5 min ($n = 4$). **b** IPGTT after 30 min ($n = 6$). **c** IPGTT after 2 h ($n = 6$). **d** IPGTT after 4 h ($n = 3$). Area under the curve (AUC) of the corresponding IPGTT curve. Data are mean ± SEM. Statistic by two-way ANOVA and Bonferroni post-test: $*p < 0.05$, $**p < 0.01$, $***p < 0.001$, comparing vehicle to oligomers; one-way ANOVA with Dunnett's multiple comparison test: $^{##}p < 0.01$, $^{###}p < 0.001$, comparing vehicle to oligomers. Source data are provided as a Source Data file

to get the same activity in vitro, they have a significant longer acting period. Interestingly, the analogue **23** with six consecutive ureido residues proved to be efficient despite its 7% potency compared to GLP-1-G$^2$ (**1**).

In order to validate our results that the pharmaceutical properties of peptide **1** can be improved by substituting amino acids with oligourea triads, we conducted a series of IPGTT studies on the three best hybrids **14**, **22** and **23** with peptide **1** as control (Fig. 4). In the first study, fasted healthy mice were submitted to an intraperitoneal glucose tolerance test (IPGTT) 5 min after dosing **1** and **14** in order to validate the activity of the reference peptide **1**. As expected both analogues were active, although **1** had a slightly higher AUC (Fig. 4a). Indeed, after 60 min of the glucose challenge, and therefore 65 min after dosing, **1** showed no more significant effect compare to the vehicle while **14** was still active, as expected from the blood glucose study (Fig. 3). In a second study, an IPGTT was performed 30 min after i.v. injections of **1**, **14** and the vehicle (Fig. 4b). An effect was observed at T0 before the glucose challenge but already after 30 min (60 min after dosing) there was no significant effect on the blood glucose for peptide **1** in agreement with previous IPGTT study. So when performing an IPGTT after 30 min of the dosing, the native peptide **1** was already inactive while hybrid **14** demonstrated full activity as predicted again from the blood glucose study. A further IPGTT study was done with a glucose challenge after 2 h of the dosing to investigate the prolonged activity of hybrids **14**, **22** and **23** (Fig. 4c). The native peptide **1** had an IPGTT curve similar to the vehicle as expected from previous studies. In contrast, the curve of hybrids **14**, **22** and **23** showed a good control of the blood glucose concentration during the IPGTT which is reflected in the AUC. An IPGTT was then performed after 4 h of dosing with compounds **1**, **14** and **22** (Fig. 4d). Almost no rise in the blood glucose was observed at 30 min post glucose challenge and the effect continued up to 120 min showing that hybrids **14** and **22** are still active after 6 h of the treatment.

To gain additional insight into the pharmacological consequences of modifying the peptide backbone with oligourea triads, we conducted a pharmacokinetic study with **1** and **22**. In good agreement with the mouse plasma study and the

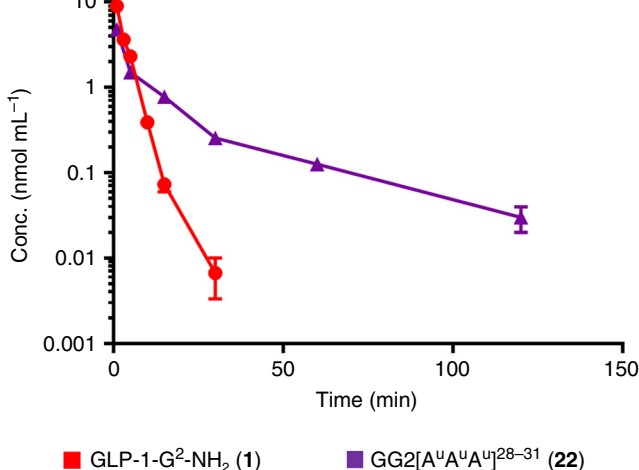

**Fig. 5** Pharmacokinetic evaluation in mice of **1** and **22** by i.v. administration. Mice (C57Bl6) treated with GLP-1 analogues (1 mg kg$^{-1}$). Data are mean ± SEM, $n = 3$. Source data are provided as a Source Data file

pharmacodynamics data, hybrid **22** had a longer half-life compared to peptide **1** (Fig. 5) presumably because it has a better proteolytic stability, although other factors might be involved. The NEP study probably indicates the trend while not being the actual enzyme that determines the in vivo half-life of our analogues. All in all, these results clearly show that the activity of peptide **1** in vivo can be prolonged using the oligoureas triads strategy by improving its pharmacodynamics and pharmacokinetics properties.

With that proof of concept in hand, we then investigated the possibility of combining the peptide/oligourea hybrid approach with the protraction strategy, which consists of functionalizing a peptide with a fatty acid chain to promote its binding to albumin and prolong its in vivo half-life. Semaglutide is a GLP-1 analogue with an Aib (2-amino isobutyric acid) in position 2 and a C18 chain linked to K20 through a small PEG spacer which was accepted by the FDA in 2018 for the treatment of diabetes as a

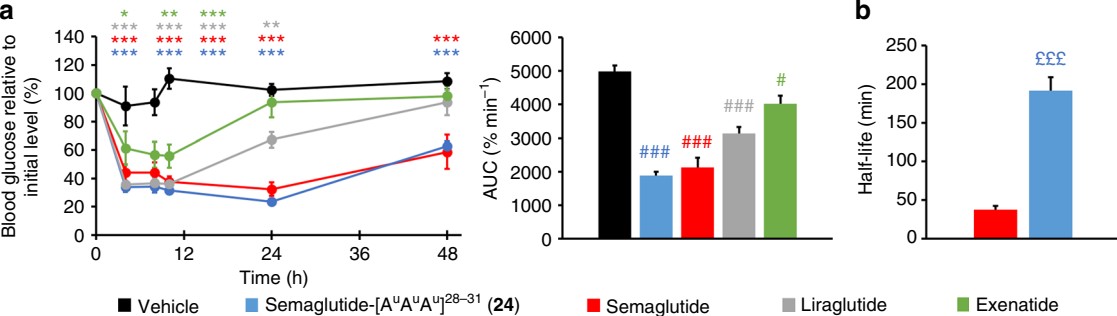

**Fig. 6** Blood glucose study in db/db mice. **a** Blood glucose concentration in db/db mice (male, 40–45 g, $n = 5$) before and after dosing: 200 μg kg$^{-1}$ (50 nmol kg$^{-1}$) intraperitoneal. Formulation: 20 μg mL$^{-1}$ in PBS 1×. The dosing was done at T0. AUC of the IPGTT curves. **b** Pancreatin degradation assay ($n = 3$). Data are mean ± SEM. Statistic by two-way ANOVA and Bonferroni post-test: *$p < 0.05$, **$p < 0.01$, ***$p < 0.001$, comparing vehicle to oligomers; one-way ANOVA with Dunnett's multiple comparison test: #$p < 0.05$, ###$p < 0.001$, comparing vehicle to oligomers; two-way t-test: £££$p < 0.001$, comparing semaglutide to hybrid **24**. Source data are provided as a Source Data file

once weekly treatment and is presently in phase 3 clinical trial for a once daily oral treatment[50,51]. We synthesized hybrid **24** which is a semaglutide analogue with an oligourea fragment replacing the four last amino acids (semaglutide-[A$^u$A$^u$A$^u$]$^{28-31}$) like in hybrid **22**. In vitro (cAMP production functional assay), hybrid **24** was found to be equally potent to semaglutide (Supplementary Fig. 5). The pharmacodynamics properties of **24** were next assessed in a study on db/db mice using FDA approved exenatide, liraglutide and semaglutide as positive controls. A single dose of analogues or placebo was injected intraperitoneal and the blood glucose was followed over time. After 48 h, hybrid **24** and semaglutide were still active while exenatide and liraglutide showed no more significant activity. Although hybrid **24** and semaglutide showed similar activities in vitro and in vivo (Fig. 6), it is noteworthy that hybrid **24** displays a longer half-life than semaglutide in mouse plasma (160 vs 11 h, Supplementary Fig. 3) and pancreatin (3.2 vs 0.63 h, Fig. 6b). Because pancreatin degradation assays are relevant to predict the stability of peptides in the gastrointestinal tract[52], our data thus suggest that hybrid **24** could be suitable for oral administration and the peptide/ oligourea hybrid approach reported here would be used to improve pharmacokinetic properties of some peptide-based drugs.

In summary, a simple approach to modify and modulate the properties of bioactive peptide helices using small foldamer inserts was developed. It consists of replacing four consecutive amino acid residues in the sequence of an α-helical peptide by an α-helicomimetic oligourea segment of three residues to generate the analogous peptide–oligourea hybrid. Here, the method was applied in a systematic fashion to the 31 amino acid peptide GLP-1 to generate a series of GLP-1-oligourea hybrids among which potent agonists of GLP-1R were identified. Agonists of GLP-1R have proved to be potent treatment against type 2 diabetes mellitus and are promising for other indications such as obesity, NASH, and Alzheimer's disease. It is noteworthy that three hybrids out of the eight tested in mice had significantly prolonged duration of action. This prolonged effect was correlated with both NEP 24.11 and mouse plasma degradation studies, suggesting an increased stabilization towards other peptidases present in the organism. The strategy was then applied to semaglutide, a FDA approved GLP-1 analogue, to generate hybrid **24** and we showed that not only the in vivo activity was preserved, but the stability toward pancreatin was improved opening the way for improvement in oral administration. Overall, this study shows that replacing four consecutives amino acid residues with an ureido triad of similar dimension and geometry in a peptide α-helix is a good strategy to improve its pharmaceutical properties. We

expect this approach to be useful for the development of peptide therapeutics and we are currently evaluating its generalization to other peptide sequences and protein targets of therapeutic interest.

## Methods

**Synthesis of GLP-1 analogues.** Compounds **1**–**24** were synthesized using solid-phase synthesis starting from Sieber amide resin (160 mg, loading 0.62 mmol g$^{-1}$). The synthesis were conducted with microwave irradiation using the Liberty Blue$^{TM}$ microwave peptide synthesizer from CEM S.A.S. N-Fmoc-α amino acid (5 equiv) were coupled with PyBOP (5 equiv) and DIEA (10 equiv) as coupling reagent using the standard Liberty Blue$^{TM}$ methods. The N-Fmoc protecting group was removed with 20% piperidine in DMF (3 mL) also with the standard methods. Each activated monomer (3 equiv) was coupled twice using DIEA (10 equiv) under microwave irradiation (70 °C, 50 W, 20 min) in DMF (4 mL). The reduction of the azido group was performed twice in a mixture of 1,4-dioxane/H$_2$O (7:3 v/v) (5 mL) with a 1 M PMe$_3$ solution in THF (10 equiv) under microwave irradiation (50 °C, 50 W, 30 min). See supplementary methods for the remaining steps of the synthesis of **24**. After completion of the synthesis, the resin was transferred into a syringe with a frit, and washed three times with DMF, three times with CH$_2$Cl$_2$ and three times with Et$_2$O. Cleavage from the resin was performed using 95% TFA with 2.5% triisopropylsilane and 2.5% water (3 mL). After 2 h the resin was filtered and discarded. Diethyl ether was added to precipitate the oligomer and the solid was triturated and filtrated. Semi preparative purification of all compound was performed by HPLC using a C18 ec column (10 × 250 mm, 5 μm).

**In vitro pharmacology (EC$_{50}$).** Evaluation of the agonist activity of compounds **1**–**23** at the mouse GLP-1 receptor endogenously expressed in βTC6 cells, was determined by measuring their effects on cAMP production using the HTRF detection method (performed by Cerep S.A., catalogue 2015, ref. 2181). Incubations of the cells were carried out in microplates at a density of $1.5 \times 10^4$ cells per well in HBSS buffer (Invitrogen) with 20 mM HEPES (pH 7.4) and 500 μM IBMX. The cells were incubated for 10 min at room temperature with HBSS (basal control), the test compound or the reference agonist and then lysed. Fluorescence donor (anti-cAMP antibody labelled with europium cryptate) and the fluorescence acceptor (D2-labelled cAMP) were then added to the mixture of lysed cells. After 60 min at room temperature, a microplate reader (Rubystar, BMG) was used to measure the fluorescence transfer at $\lambda_{ex} = 337$ nm and $\lambda_{em} = 620$ and 665 nm. The ratio of the signal measured at 665 nm on signal measured at 620 nm is used to determine the cAMP concentration. The results are given as a percent of the control response to 10 nM GLP-1. The standard reference agonist is GLP-1, which is tested in each experiment at several concentrations to generate a concentration-response curve from which its EC$_{50}$ value and SE is calculated using GraphPad Prism.

**Enzymatic degradation (NEP 24.11).** Stock solutions of the oligomers were prepared at a concentration of 400 μM in a solution of 50 mM HEPES buffer, 50 mM NaCl, 0.05% Tween-80, pH 8.0. Stock solution of NEP 24.11 was prepared at a concentration of 100 μg mL$^{-1}$ in water. Stability of oligomers to NEP 24.11 was assessed by conducting protease reaction in a 96-well plates at 20 °C. To each well was added 2 μL of HEPES buffer, 38 μL of the solution of oligomer to be assayed (final concentration 304 μM) and 10 μL of the solution of enzyme (final concentration 20 μg mL$^{-1}$) for a total volume of 50 μL. Each compound was also incubated in the absence of the enzyme (12 μL of HEPES buffer and 38 μL of oligomer). At the indicated time (1, 2, 4, 6 and 22 h) an aliquot of 10 μL was removed from each experimental reaction and pipetting into 100 μL of 1% TFA

solution to quench the reaction ($t = 0$ min was determined using the reaction without enzyme). A portion of the quenched reaction solution was analysed by HPLC. The time course of peptide degradation was determined by integrating the area of each peak in a series of HPLC traces.

**Enzymatic degradation (pancreatin)**. Stock solutions of the oligomers were prepared at a concentration of 250 μM in DMSO. Stock solution of pancreatin was prepared at a concentration of 10 mg mL$^{-1}$ in water. Two microlitres of pancreatin stock solution was diluted 1/500 with a solution of TRIS 10 mM pH 7.5 to afford a final concentration of 0.02 mg mL$^{-1}$. The oligomer was then diluted 1/24 with a solution of TRIS 10 mM pH 7.5 to afford a final concentration of 10 μM and incubated at 37 °C (8.3 μL of the oligomer stock solution was diluted with 190 μL of TRIS 10 mM pH 7.5 and 1.7 μL of the pancreatin solution). Each compound was also incubated in the absence of pancreatin (8.3 μL of the oligomer stock solution was diluted with 190 μL of TRIS 10 mM pH 7.5). At the indicated time (20 min and 60 min), an aliquot of 70 μL was removed from each experimental reaction and pipetting into 175 μL of acetonitrile at 0 °C to quench the reaction ($t = 0$ min was determined using the reaction without pancreatin). The samples were frozen at –80 °C before analysis. The frozen samples were defrosted, stirred with a vortex 5 min and finally centrifuged 5 min at 16 °C ($1500 \times g$). The supernatant was analysed by LC-MS. The time course of oligomer degradation was determined by integrating the area of the peak in the extracted ion chromatogram.

**Mouse plasma stability**. Stock solutions of the oligomers were prepared at a concentration of 250 μM in water. The oligomer was then diluted 1/50 with a solution of plasma/PBS pH 7.4 (1:1) to afford a final concentration of 5 μM and incubated at 37 °C (4 μL of the stock solution was diluted with 196 μL of plasma/PBS, pH 7.4, 1:1). Each compound was also incubated in the absence of plasma (196 μL of H$_2$O/PBS, pH 7.4, 1:1). At the indicated time (20 min and 60 min) an aliquot of 70 μL was removed from each experimental reaction and pipetting into 175 μL of acetonitrile at 0 °C to quench the reaction ($t = 0$ min was determined using the reaction without plasma). The samples were frozen at –80 °C before analysis. The frozen samples were defrosted, stirred with a vortex 5 min and finally centrifuged 5 min at 16 °C ($1500 \times g$). The supernatant was analysed by LC-MS. The time course of oligomer degradation was determined by integrating the area of the peak in the extracted ion chromatogram.

**Animals**. For the pharmacodynamics studies performed by Physiogenex S.A.S., mice were housed in ventilated and enriched housing cages ($310 \times 125 \times 127$ mm³) throughout the experimental phase. The mice were housed in groups of three animals during the study, on a normal 12 h light cycle (at 8:00 pm lights off), 22 ± 2 °C and 50 ± 10% relative humidity. A standard chow diet (RM1 (E) 801492, SDS) and tap water were provided ad libitum. All animal protocols done by Physiogenex S.A.S. were reviewed and approved by the local (Comité régional d'éthique de Midi-Pyrénées) and national (Ministère de l'Enseignement Supérieur et de la Recherche) ethics committees (protocol number 05049–06). For the pharmacokinetics studies performed by TechMedILL service (PCBIS platform, CNRS UMS3286), mice were housed in polycarbonate cages (PCT2L12SHT, Allentown) enriched with play tunnels throughout the experimental phase. The mice were housed in groups of nine animals during the study, under controlled environment (22 ± 1 °C) with a relative humidity (50 ± 10%) and a normal 12 h light cycle (at 8:00 pm lights off). A standard chow diet (A04, SAFE, France) and tap water were provided ad libitum. All animal protocols done by TechMedILL service were reviewed and approved by the agriculture ministry regulating animal research in France (Ethics regional committee for animal experimentation Strasbourg, APAFIS 1341#2015080309399690).

**Blood glucose experiment in healthy mice**. After the acclimation period (at least 5 days), mice (male C57BL/6J mice (Charles River laboratories) 8 weeks old, 20–25 g) were randomized into 10 groups ($n = 3$ per group) according to their body weight. They were acutely treated via i.v. route at 10 am (5 μg per mouse). Blood glucose was measured before dosing and at 30 min, 1, 2, 4, 6 h after dosing in fed conditions.

**IPGTT experiments in mice**. After the acclimation period (at least 5 days), mice (male C57BL/6J mice (Charles River laboratories) 6 weeks old, 20–25 g) were randomized into groups according to their body weight. The mice were fasted for 6 h prior being acutely treated via i.v. route (5 μg per mouse i.v. (200 μg kg$^{-1}$, 50 nmol kg$^{-1}$)) formulated at 20 μg mL$^{-1}$ in PBS 1×. The IPGTTs were performed (glucose 2 g kg$^{-1}$ i.p.) 5 min, 30 min, 2 h or 4 h after dosing and blood glucose was measured at different time points after the glucose challenge.

**Blood glucose experiment in db/db mice**. After the acclimation period (at least 5 days), mice (male db/db mice (Charles River laboratories) 7 weeks old, 40 g) were fasted for 6 h, then blood was collected to measure levels of glucose and insulin. Mice were then randomized in five homogenous groups ($n = 5$ per group) according to their blood glucose, HOMA-IR and body weight. Then, mice were refed until the day of treatment (at least 2 weeks). At the treatment day, mice

received a single intraperitoneal injection of test items (200 μg kg$^{-1}$, 50 nmol kg$^{-1}$) formulated at 20 μg mL$^{-1}$ in PBS 1× or vehicle. Blood was collected from the tail vein and blood glucose levels was measured before the injection and 4, 8, 10, 24 and 48 h after dosing in fed conditions.

**Pharmacokinetics**. Fifteen mice (male C57BL/6J mice (Janvier Labs, France) 9 weeks old, 20–25 g) were treated with GLP-1 analogues via i.v. injections (2 mg kg$^{-1}$) formulated at 1 mg mL$^{-1}$ in PBS 1×. After different time points, mice were sacrificed and blood samples were collected. The plasma was separated by centrifugation and the samples were frozen at −80 °C before analysis. A volume of 400 μL of each sample of plasma was mixed with 1 ml of acetonitrile to precipitate the protein and extract the compound. The sample were then vortexed and centrifuged ($15,000 \times g$, 5 min, 16 °C) to sediment the precipitated protein. The supernatant was analysed by LC-MS/MS using a UHPLC coupled to LC-MS 8030 Shimadzu triple quadrupole.

**Statistical analysis**. GraphPad Prism 7.0 was used for all analyses. Replicate measurements were taken from different samples. Statistical analyses (two-way t-test, two-way ANOVA and Bonferroni post-test and one-way ANOVA with Dunnett's multiple comparison test) were applied when indicated. P values lower than 0.05 were considered significant.

**Reporting summary**. Further information on experimental design is available in the Nature Research Reporting Summary linked to this article.

## Data availability

Data supporting findings of this manuscript are available from the corresponding authors upon reasonable request. Additional information on the synthesis of the GLP-1 analogues and their characterizations is available in the Supplementary data (Supplementary Methods and Supplementary Figs. 7–87). The source data underlying Figs. 2–7 and Supplementary Figs. 1–6 are provided as a Source Data file.

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

## Acknowledgements

This work was supported by UREKA Sarl, Immupharma Group. Support from the Conseil Regional de Nouvelle-Aquitaine (Project #14004308), Univ. Bordeaux and ANR (Grant ANR-15-CE07-0010) is also gratefully acknowledged. S.R.G. is grateful to Thierry Sulpice and Nourdine Faresse (Physiogenex S.A.S.) for insightful discussions. This work has benefited from the facilities and expertise of IECB Biophysical and Structural Chemistry platform (BPCS), CNRS UMS3033, Inserm US001, Univ. Bordeaux.

## Author contributions

J.F., C.V., R.H.Z., G.G. and S.R.G. designed the analogues. J.F. and C.V. synthesized the analogues and analysed results. J.F. and L.M. performed the NEP 24.11, mouse plasma and pancreatin degradation assays and analysed results. J.F., C.V., R.H.Z., L.M., G.G. and S.R.G. interpreted in vitro pharmacology and in vivo studies results. G.G. and S.R.G. conceived the project. S.R.G. supervised the project. G.G. and S.R.G. analysed the data and wrote the manuscript.

## Additional information

**Competing interests:** J.F., R.H.Z. and G.G. are inventors on a patent application covering GLP-1 analogues described here. J.F., C.V., L.M., R.H.Z. and S.R.G. work for UREKA Sarl, which is pursuing biomedical applications of peptide–oligourea hybrids. L.M., R.H.Z., G.G. and S.R.G. are shareholders of ImmuPharma PLC, the parent company of UREKA Sarl.

