## [Peer Review File · Nature Communications]

Reviewers' comments:

Reviewer #1 (Remarks to the Author):

It is too soon to publish this paper. In future, may make a good paper. Interesting observations, but not enough explanations.

Authors describe that analogs of GLP-1 hormone with ureas in backbone can be agonists of GLP-1 receptor. Not all urea locations are active, but some are. Authors ask if urea insertions prevent cutting by neprilysin. Data in table 2. All analogs cut about as fast as GLP-1 (really, a small difference from GLP-1, peptide 1). Authors try to interpret small differences in $t_{1/2}$, but really all analogs more or less the same. These results could be predicted because neprilysin cuts in many sites along GLP-1.

Strange results in mice. Figure 4e shows that GLP-1-like peptide 1 is not active in IPGT test. Why is this? The peptide should be active at the concentration used. If this positive control doesn't work, hard to know what to say about rest of the data.

Authors describe that a few urea analogs of GLP-1 are active in mice, and active for a long time. But long activity is not predicted by neprilysin results. So why the long activity? Authors need to try to answer this question (and explain about failed positive control) before publishing. Are long-active analogs pretty hydrophobic? Authors should know from HPLC retention comparisons. Maybe these analogs form depot or stick to blood proteins? Authors should do some PK studies to see if active analogs stay in blood a long time.

Reviewer #2 (Remarks to the Author):

The manuscript by Fremaux et al. describes the synthesis and biological evaluation of α -peptide-oligourea hybrid analogues of the peptide hormone GLP-1. In the design of those analogues, four consecutive amino acids from the GLP-1 sequence are replaced by three ureido residues. Next, an Ala u scan is performed, giving rise to more than 20 analogues which are then evaluated. Many of them show potency in vitro, as well as an improved half-life, and for some of them good results are also observed in vivo.

Although the synthetic methodology for the peptide-oligourea hybrids has been developed by the authors in previous publications, it is noteworthy the degree of α -helical mimicry accomplished by the ureido triad inserts designed in this work. In addition, the prolonged half-life observed in vivo for some of the analogues studied, opens the door to use this strategy in other bioactive peptides showing proteolytic degradation, since this is still one of the main limitations in peptide therapeutics. I therefore, think that this work deserves publication in Nature Communications.

The authors should, however, revise the manuscript for typographic and gramatical errors. As an example, the following sentence should be revised:

'This represent the first reported peptide-oligourea hybrids as α -helix mimics with improve action in vivo.'

Reviewer #3 (Remarks to the Author):

In this manuscript, authors described a strategy of peptide-oligourea hybrids mimicking α -helix, and employed this strategy to modify GLP-1 from Asp9 to Gly31. They found that three analogues of GLP-1 show improved duration of action. The strategy for development of GLP-1 analogues is new. However, since several stabilized GLP-1 analogues are on the market or in clinical trials, I think that the trend of this field is to find more long-term GLP-1R agonists or oral therapeutics. In this manuscript, although some modified GLP-1 analogues showed some improved activities, they

do not have obvious advantages over the existing GLP-1 analogues that are on the market. Therefore, in my opinion, this manuscript cannot be published in this high-level journal.

Some revisions are needed:

At least, data of longer term (24 h even 48 h) glucose-lowering effects of active analogues should be included.

Reviewers' Comments (plus author responses):

Reviewer #1 (Remarks to the Author):

It is too soon to publish this paper. In future, may make a good paper. Interesting observations, but not enough explanations.

We are grateful to the reviewer for acknowledging the potential of our work and for making very useful suggestions. The revised manuscript contains new results (mouse plasma degradation studies, a PK study and new IPGTT experiments) which contribute to further explain the biological effects of the oligourea triad inserts. Overall, we now believe that the concerns expressed by this reviewer have been addressed, and that the quality of the manuscript has improved substantially.

Point 1: Authors describe that analogs of GLP-1 hormone with ureas in backbone can be agonists of GLP-1 receptor. Not all urea locations are active, but some are. Authors ask if urea insertions prevent cutting by neprilysin. Data in table 2. All analogs cut about as fast as GLP-1 (really, a small difference from GLP-1, peptide 1). Authors try to interpret small differences in $t_{1/2}$, but really all analogs more or less the same. These results could be predicted because neprilysin cuts in many sites along GLP-1. This is a good point raised by the reviewer which has now been addressed.

In order to verify if the differences observed in the NEP 24.11 study were significant, we conducted statistical analyses on the data on both the $t_{1/2}$ and the % remaining at different time points. A two-way anova with Bonferroni post-test on the different time points showed that five compounds were significantly different than the native peptide 1 when you look at the 4 h and 6 h points (Fig. R1 and Fig. S76). The statistical analyses at different time points therefore allows to discriminate the analogues from peptide 1. It is worth noting that this is independent from the method chosen to calculate the half-lives.

Figure R1. NEP 24.11 Study

To make the half-lives values more representative of the results at the 4 h and 6 h time point, we also decided to slightly modify the non-linear regression method to determine the $t_{1/2}$: we fixed the plateau at 0% in the regression, which was not done in the previous version of the article. The NEP 24.11 $t_{1/2}$ values were therefore slightly modified from the last version. Although only 23 had a significantly different $t_{1/2}$ in the NEP 24.11 degradation assays, we can still conclude from the

different time points that five compounds were significantly different from the native peptide. These new results (NEP 24.11 $t_{1/2}$) have been added in Table 2 of the revised manuscript.

Furthermore, we modified the text to discuss these new data:

*“The half-lives of compounds **1**, **2**, **5**, **9**, **11**, **14**, **16**, **22** and **23** in presence of NEP 24.11 and in presence of mouse plasma were determined by following the decay of the analogues by LCMS (Table 2). In the NEP 24.11 degradation assay, the native peptide **1** had a half-life of 3.6 h and unfortunately the hybrids **5**, **9** and **16** showed diminished half-lives of 2.9 h, 3.0 h and 1.8 h respectively. Although these $t_{1/2}$ values were not significantly different from the $t_{1/2}$ of peptide **1** (3.6 h), statistical significance was obtained at the 4 h time point for these three hybrids demonstrating their lower stability (Supplementary data). Those results were intriguing, as we did not expect lower proteolytic protection for some hybrids. Further investigations showed no indication that the cleavage sites for the destabilized hybrids were located in or next to the oligourea triads making it more puzzling. Nevertheless, in two cases (**22**, **23**) the permutation of 4 amino acids for an oligourea triad improved significantly the proteolytic protection of the hybrids (Table 2 and Supplementary data). It is also noteworthy that the analogue **23** where 8 amino acids were replaced by 2 triads had a significant increase of its half-life.”*

Strange results in mice. Figure 4e shows that GLP-1-like peptide **1** is not active in IPGT test. Why is this? The peptide should be active at the concentration used. If this positive control doesn't work, hard to know what to say about rest of the data.

The finding that the GLP-1 like peptide **1** (GLP-1-G²) is not active in IPGTT experiments when the glucose challenge is performed 2h after dosing is not surprising given the susceptibility of GLP-1 to proteolytic degradation. It is true that it has been published that GLP-1 is still active after 1 h (i.e. Johnson, L. M.; et al. *J. Am. Chem Soc.* 2014, 136 (37), 12848–12851.), however in all the examples we found, GLP-1 was given via the i.p. route, while we made i.v. administrations.

Nevertheless, we undertook two new studies to address this interrogation. IPGTT studies with glucose challenge at 5 min and 30 min after dosing were conducted to demonstrate the activity of peptide **1**. As expected, peptide **1** was active when performing the IPGTT after 5 min, but not after 30 min. These results are corroborated by the new mouse plasma degradation study and the pharmacokinetics study (vide infra).

In the text we added this:

*“In order to validate our results that the pharmaceutical properties of peptide **1** can be improved by substituting amino acids with oligourea triads, we conducted a series of IPGTT studies on the three best hybrids **14**, **22** and **23** with peptide **1** as control. In the first study, fasted healthy mice were submitted to an intra peritoneal glucose tolerance test (IPGTT) 5 min after dosing **1** and **14** in order to validate the activity of the reference peptide **1**. As expected both analogues were active, although **1** had a slightly higher AUC (Figure 5-A). Indeed, after 60 min of the glucose challenge, and therefore 65 min after dosing, **1** showed no more significant effect compare to the vehicle while **14** was still active, as expected from the blood glucose study (Figure 4). In a second study, an IPGTT was performed 30 min after i.v. injections of **1**, **14** and the vehicle. An effect was observed at T0 before the glucose challenge but already after 30 min (60 min after dosing) there was no significant effect on the blood glucose in agreement with previous IPGTT study (Figure 5-B1). So when performing an IPGTT*

after 30 min of the dosing, the native peptide 1 was already inactive while hybrid 14 demonstrated full activity as predicted again from the blood glucose study.”

It is worth noting that in four in vivo studies (Fig. 4, Fig. 5A1 and 5B1, Fig. S78) we observed consistently the same result for peptide 1: an effect at 30 min and no more effect at 60 min post dosing. In addition to proving the effect of GLP-1-G² (1), these new experiments demonstrate even more to which extend we ameliorated peptide 1 by replacing four amino acids with an oligourea triad (e.g. compounds 14, 22).

Point 2: Authors describe that a few urea analogs of GLP-1 are active in mice, and active for a long time. But long activity is not predicted by neprilysin results. So why the long activity? Authors need to try to answer this question (and explain about failed positive control) before publishing.

This is indeed an important point raised by the reviewer. As mentioned above, the revisited analysis of the data of NEP 24.11 degradation assay suggests that the combination of NEP 24.11 and in vitro agonist assays can reasonably predict the in vivo data.

However, in order to provide stronger evidence of an increased proteolysis protection for some of the hybrids, we conducted a series of mouse plasma degradation assays. The results we obtained were more compelling and interestingly they were mostly predicted by the NEP 24.11 study, thus providing a strong support to the main conclusions of the manuscript.

These new results (Mouse plasma $t_{1/2}$) have been added in Table 2 of the revised manuscript as shown below:

Compound	Potency (%) ^a	NEP 24.11 $t_{1/2}$ (h) ^b	SEM (h) ^c	Mouse Plasma $t_{1/2}$ (min) ^d	SEM (min) ^e	In vivo activity ^f	
1	GLP-1-G ² -NH ₂ (GG2)	100	3.6	0.1	16	1	+
2	GG2[D ^u A ^u A ^u] ⁹⁻¹²	2	4.4	0.4	17	1	
5	GG2[A ^u Y ^u A ^u] ¹¹⁻¹⁴	15	2.9	0.3	23	3	+
9	GG2[Y ^u E ^u A ^u] ¹⁴⁻¹⁷	135	3.0	0.2	17	1	+
11	GG2[E ^u A ^u A ^u] ¹⁵⁻¹⁸	13	3.9	0.4	31*	3	
14	GG2[A ^u A ^u A ^u] ¹⁸⁻²¹	6	4.9	0.8	38***	2	+++
16	GG2[F ^u A ^u A ^u] ²²⁻²⁵	0.4	1.8	0.1	38***	5	
22	GG2[A ^u A ^u A ^u] ²⁸⁻³¹	19	4.5	0.5	>60***	na	+++
23	GG2[Y ^u E ^u A ^u A ^u A ^u] ¹⁴⁻²¹	7	6.1*	0.9	39***	5	+++

^a Percentage of potency (cAMP production) compared to GLP-1-G² (see Table 1). ^b Half-life of the GLP-1 analogues in NEP 24.11 degradation assay: mean value of three replicates. ^c Standard error of the mean on the NEP 24.11 $t_{1/2}$. ^d Half-life of the GLP-1 analogues in mouse plasma degradation assay: mean value of four replicates. ^e Standard error of the mean on the mouse plasma $t_{1/2}$. ^f Indicator of *in vivo* activity. + = active, +++ = more active. (one way anova with Dunnett's multiple comparison test: * p<0,05; *** p<0,001)

Furthermore, we modified the text to discuss these new data:

“ In mouse plasma degradation assay, the native peptide 1 had a half-life of 16 min and hybrids 2, 5 and 9 had similar $t_{1/2}$ of 17, 23 and 17 min respectively. Interestingly, hybrids 11, 14, 16, 22, and 23 had significantly longer half-lives. In the case of 22, a half-life of >60 min was observed, which represents over 3.8-folds improvement compared to peptide 1. Over the 8 hybrids selected, 5 had improved stability in mouse plasma with the insertion of the triad at various positions on the peptide. The proteolytic stability to NEP 24.11 could predict mostly the results in mouse plasma. The next step was to investigate if these improvements in half-lives could be transposed in vivo.”

Point 3: Are long-active analogs pretty hydrophobic? Authors should know from HPLC retention comparisons. Maybe these analogs form depot or stick to blood proteins?

These are interesting suggestions but we believe that the new data and discussion we provided in response to points 1 and 2 of the same reviewer (*vide supra*) are fully consistent with the prolonged activity reported for some of the ureido modified GLP-1 analogues. Our current understanding is that there are multiple factors that are in cause for the clearance of the GLP-1 analogues, and in order to improve their pharmaceutical properties, we have to improve the peptide against many proteases and enzymes as well as limiting the elimination by filtration of the kidneys. Although the change in lipophilicity might have a role to play, it cannot explain all the result.

When we compare the plasma $t_{1/2}$ with the HPLC retention times, the data can be analysed in two ways (See Figure below for reviewing only). When you include **22**, the data fits an exponential growth curve ($R^2 = 0.97$). However, if you exclude **22**, you can see there is no significant correlation ($R^2 = 0.36$) between the mouse plasma $t_{1/2}$ and the HPLC retention time. For compound **22**, its lipophilicity might have a higher role to play in its improved properties, however this simple parameter could not explain the results for **14** and **23**.

Authors should do some PK studies to see if active analogs stay in blood a long time.

This is again an excellent suggestion which we have taken on board as the new analogues can display specific pharmacokinetic profiles. A pharmacokinetics study was conducted on the native peptide **1** and the analogue with the longest half-life in plasma (**22**) and the results are summarized below. As expected, peptide **1** disappeared very rapidly when injected by i.v. and hybrid **22** had a clearance rate 20-folds lower. The pharmacokinetics profiles of **1** and **22** corresponded to what was expected from the pharmacodynamics profiles (Blood glucose study and IPGTT studies).

Table 3 | Pharmacokinetic evaluation in mice by i.v. administration

		Cl (mL/min/kg)	Vz (mL/kg)	$t_{1/2}$ (min)
1	GLP-1-G ² -NH ₂	448	4484	7
22	GG2[A ^u A ^u A ^u] ²⁸⁻³¹	22	711	23

^a Pharmacokinetic in normal mice (C57BL/6J, male, 20-25 g, n = 3) dosing: 25 µg/mouse (~250 nmol/kg) intra venous. Formulation: 2 mg/mL in PBS 1X.

An additional table has been prepared (Table 3) and the following text was added:

*“A pharmacokinetic study was also conducted in mice to access the pharmacologic properties of **1** and **22**. As expected by the mouse plasma studies and the pharmacodynamics studies, hybrids **22** had a longer $t_{1/2}$, a lower clearance and a lower Vz compared to the native peptide **1**. Those results clearly show that the activity of peptide **1** in vivo can be prolonged using the oligourea triads strategy by improving its pharmacodynamics and pharmacokinetics properties.”*

Reviewer #2 (Remarks to the Author):

The manuscript by Fremaux et al. describes the synthesis and biological evaluation of α -peptide-oligourea hybrid analogues of the peptide hormone GLP-1. In the design of those analogues, four consecutive amino acids from the GLP-1 sequence are replaced by three ureido residues. Next, an Ala u scan is performed, giving rise to more than 20 analogues which are then evaluated. Many of them show potency in vitro, as well as an improved half-life, and for some of them good results are also observed in vivo.

Although the synthetic methodology for the peptide-oligourea hybrids has been developed by the authors in previous publications, it is noteworthy the degree of α -helical mimicry accomplished by the ureido triad inserts designed in this work. In addition, the prolonged half-life observed in vivo for some of the analogues studied, opens the door to use this strategy in other bioactive peptides showing proteolytic degradation, since this is still one of the main limitations in peptide therapeutics. I therefore, think that this work deserves publication in Nature Communications.

We thank the reviewer for his positive comments and for acknowledging that our work “opens the door to use this strategy in other bioactive peptides” and therefore has a general application in both peptide chemistry and medicinal chemistry.

The authors should, however, revise the manuscript for typographic and gramatical errors. As an example, the following sentence should be revised:

'This represent the first reported peptide-oligourea hybrids as α -helix mimics with improve action in vivo.'

The manuscript was revised for typographic and grammatical errors and the sentence above was modified for:

“This represents the first reported block peptide-oligomer hybrid with improved action in vivo in which a non- α -amino-acid-containing oligomer is used as an α -helix mimic.”

Reviewer #3 (Remarks to the Author):

In this manuscript, authors described a strategy of peptide-oligourea hybrids mimicking α -helix, and employed this strategy to modify GLP-1 from Asp9 to Gly31. They found that three analogues of GLP-1 show improved duration of action. The strategy for development of GLP-1 analogues is new. However, since several stabilized GLP-1 analogues are on the market or in clinical trials, I think that the trend of this field is to find more long-term GLP-1R agonists or oral therapeutics. In this manuscript, although some modified GLP-1 analogues showed some improved activities, they do not have obvious advantages over the existing GLP-1 analogues that are on the market. Therefore, in my opinion, this manuscript cannot be published in this high-level journal.

Some revisions are needed:

At least, data of longer term (24 h even 48 h) glucose-lowering effects of active analogues should be included.

The point this reviewer makes raises an issue which, although interesting, we felt did not fit within the scope of this manuscript. We did not intend to suggest that the compounds we report are direct competitors of GLP-1 analogues on the market or already in clinical trials – our paper is motivated by a desire to describe and evaluate a new methodology based on original backbone modifications to mimic and augment pharmaceutical properties of bioactive peptides. Our primary goal was to demonstrate the efficacy of the approach and to show how it can be applied. Although GLP-1R is a relevant target and has still a lot of market potential, GLP-1 was used primarily as a showcase of the technology. The revised manuscript contains considerable new data to address this general issue. As stated by this reviewer this approach is new. Sequence-based biopolymer mimicry with foldamers has proven particularly challenging and only few suitable backbones (e.g. α/β -peptides introduced by Gellman) have been shown to provide suitable mimics. The chiral oligourea foldamers used in this study combine a number of unique and positive features – such as synthetic accessibility, sequence modularity, and high folding fidelity – which could be further exploited to turn this concept into a practical and scalable technology.

It is worth noting that the development of peptide therapeutics generally requires the use of multiple strategies in order to obtain the optimal pharmacological properties. The approach developed in this study can certainly be combined with other known methods such as side chain bridging (i.e. so called stapled peptides), the addition of a fatty side chain, addition of a PEG, fusion to large proteins, etc. Particularly, the use of lipidation or fusion to large protein have been successful in the case of GLP-1 when combined with other strategies to prevent proteolysis. Although these strategies could be combined with the urea-based GLP-1 analogues described here to generate long lasting GLP-1 analogues, we would respectfully argue that this was beyond the scope of this study for the reasons mentioned above.

Nevertheless, we now provide data from new experiments showing that compounds **14** and **22** are still active more than 4 hours after injection. This is remarkable knowing that under the same conditions the cognate peptide loses its activity after 30 minutes. These new results have now been included in Figure 5D1 and 5D2 and the following text was added to the main text :

*“An IPGTT was then performed after 4 h of dosing with compounds **1**, **14** and **22**. Almost no rise in the blood glucose was observed at 30 min post glucose challenge and the effect continued up to 120 min showing that hybrids **14** and **22** are still active after 6 h of the treatment (Figure 5-D1).”*

Reviewers' comments:

Reviewer #1 (Remarks to the Author):

Authors have done extra experiments, and manuscript is stronger than before. But the most important questions are not answered.

Before I said that most analogs are not very different than GLP-1 G2 in neprilysin cutting. Now the authors have analyzed very small differences in a lot of detail, but general conclusion is not changed. For example, peptide 22 half-life in presence of neprilysin is less than 50% longer than half-life of GLP-1 G2 (Table 2), but peptide 22 is active for a longer time in vivo. So you have to conclude that the long in vivo activity of peptide 22 is not related to proteolysis by neprilysin. But the authors do not make this conclusion. They seem to ignore this conclusion. There are other ways to see the puzzle of long action of 22 in vivo. How can you explain that the $t_{1/2}$ in mice is only 3-fold different for GLP-1 G2 vs 22, but the clearance is 20-fold different? How can you explain that GLP-2 G2 is inactive in vivo after four half-lives, but 22 is active after more than 10 half-lives? The in vivo activity of analogs like 22 is interesting, but authors do not seem to realize the aspects that are most interesting.

This is ok for Nature Communications as an observation about an unusual kind of peptide modification. That observation will get attention from many readers. But the paper is puzzling about the reasons, and the authors do not seem to recognize the puzzle.

Reviewer #2 (Remarks to the Author):

In my previous report I already recommended publication of the manuscript by Fremaux et al. after minor revisions, which have been addressed. Now with the additional experiments, I think the manuscript is much improved and the conclusions are well supported. I therefore recommend publication in Nature Communications.

Reviewer #3 (Remarks to the Author):

α -Peptide-oligourea chimeras which can act as peptide helical foldamers was developed by Dr. Gilles Guichard (Angew. Chem., 2015, 127, 9954–9958). In this manuscript, the authors applied this strategy to GLP-1 modification. It is a concept extension.

In this revised manuscript, the authors added the new experiments which partially address my concerns. This manuscript might be publishable in this journal on condition that some new experiments need to be performed. I still suggest that at least 24 h glucose-lowering effects of active analogues should be included.

Reviewers' Comments (plus author responses):

Reviewer #1 (Remarks to the Author):

Authors have done extra experiments, and manuscript is stronger than before.

We are grateful to the reviewer for acknowledging that our work "is stronger than before".

But the most important questions are not answered.

Before I said that most analogs are not very different than GLP-1 G2 in neprilysin cutting. Now the authors have analyzed very small differences in a lot of detail, but general conclusion is not changed. For example, peptide 22 half-life in presence of neprilysin is less than 50% longer than half-life of GLP-1 G2 (Table 2), but peptide 22 is active for a longer time in vivo. So you have to conclude that the long in vivo activity of peptide 22 is not related to proteolysis by neprilysin. But the authors do not make this conclusion. They seem to ignore this conclusion.

We are thank the reviewer for these additional comments. We agree that neprilysin might not play a key role in the proteolysis of our analogues, however we cannot rule this out.

We added this text:

"The NEP study probably indicates the trend while not being the actual enzyme that determines the in vivo half-life of our analogues."

There are other ways to see the puzzle of long action of 22 in vivo. How can you explain that the t1/2 in mice is only 3-fold different for GLP-1 G2 vs 22, but the clearance is 20-fold different? How can you explain that GLP-2 G2 is inactive in vivo after four half-lives, but 22 is active after more than 10 half-lives? The in vivo activity of analogs like 22 is interesting, but authors do not seem to realize the aspects that are most interesting.

We agree with the reviewer that our pharmacokinetic data were somewhat puzzling. We hypothesized that this apparent discrepancy between in vivo activity and pharmacokinetics (PK) and difficulty to draw firm conclusions could be due to the experimental settings and more specifically to: 1) the low amount of material injected in the mice and 2) the absence of time points closer to the time of injection. To strengthen our PK data, we have decided to conduct another PK study by doubling the quantity injected and measure product concentration in blood at additional time points (1 min, 3 min, 5 min). The new PK results are shown in new Fig 6. Although only analyzed in a qualitative manner, the results clearly show the much longer half-life of compound **22** as compared to **1**. The following text was added:

*"To gain additional insight into the pharmacological consequences of modifying the peptide backbone with oligourea triads, we conducted a pharmacokinetic study with **1** and **22**. In good agreement with the mouse plasma study and the pharmacodynamics data, hybrid **22** had a longer half-life compared to peptide **1** (Figure 6) presumably because it has a better proteolytic stability, although other factors might be involved. The NEP study probably indicates the trend while not being the actual enzyme that determines the in vivo half-life of our analogues. All in all, these results clearly show that the activity of peptide **1** in vivo can be*

prolonged using the oligoureas triads strategy by improving its pharmacodynamics and pharmacokinetics properties.”

This is ok for Nature Communications as an observation about an unusual kind of peptide modification. That observation will get attention from many readers. But the paper is puzzling about the reasons, and the authors do not seem to recognize the puzzle.

We are grateful to the reviewer for acknowledging that our work is “ok for Nature Communications” and it “will get attention from many readers”. We think that the new PK study together with the design and study of a new compound (i.e. a semaglutide analogue containing a oligourea triad), provide additional insight on how the oligourea insert may influence the pharmaceutical properties of peptides (see also response to reviewer 3).

Reviewer #2 (Remarks to the Author):

In my previous report I already recommended publication of the manuscript by Fremaux et al. after minor revisions, which have been addressed. Now with the additional experiments, I think the manuscript is much improved and the conclusions are well supported. I therefore recommend publication in Nature Communications.

We thank the reviewer for his positive comments and for acknowledging that our manuscript “is much improved and the conclusions are well supported”.

Reviewer #3 (Remarks to the Author):

α -Peptide–oligourea chimeras which can act as peptide helical foldamers was developed by Dr. Gilles Guichard (Angew. Chem., 2015, 127, 9954–9958). In this manuscript, the authors applied this strategy to GLP-1 modification. It is a concept extension.

We respectfully disagree with the statement that our work is a concept extension. In the earlier work published in Angew. Chem., 2015, 127, 9954–9958 to which the reviewer refers, we showed that peptide/oligoureas hybrids adopt a well-defined helical conformation in organic solution with the two helix types (i.e. the α -helix and oligourea helix) coexisting and communicating through a specific H-bonding network. Another trend from this study was that short oligourea caps could be sufficient to stabilize a helical conformation. Although these data still need to be validated in aqueous environment, this key discovery encouraged us to examine whether the α -helical backbone could be replaced by such oligourea mimics at specific locations in biologically active peptides and what would be the consequences on the biological activities. However, by no mean, it was possible to predict the outcome: forming a helix is one thing, preserving binding and affinity to the target surface and maintaining the overall positioning of all the side chains is another. Only few foldamer systems have actually been reported in the literature with a backbone that effectively mimic the α -helical backbone for inhibiting protein-protein interactions or activating receptors. Not only did we show that it was possible to make active analogues, we also showed that these analogues display improved pharmaceutical properties and this result was not anticipated considering the importance of backbone modification introduced.

In this revised manuscript, the authors added the new experiments which partially address my concerns. This manuscript might be publishable in this journal on condition that some new experiments need to be performed. I still suggest that at least 24 h glucose-lowering effects of active analogues should be included.

Again, the main point of this article was not about making long lasting GLP-1 analogues, but to demonstrate that peptide/oligourea hybrids can be added in the short list of efficient methods to improve the pharmaceutical properties of a peptide. Nevertheless, we have now considered the request of the reviewer by introducing the oligourea insert reported for compound **22** in the backbone of a clinically approved GLP-1R agonist (i.e. Semaglutide). Thus, a new analogue was prepared (**24**), characterized and evaluated *in vitro* in a cAMP production functional assay and *in vivo* in *db/db* mice. Results show that this compound, has a glucose-lowering effect longer than 48 h similar to that of semaglutide. Interestingly the compound also manifests increased resistance to degradation by pancreatin which is an advantage in the context of oral drug delivery. The manuscript has been changed to accommodate these new results by adding one Figure (i.e. new Figure 7), plus additional content as supplementary data (*in vitro* analysis) as well as an additional paragraph as follow:

*“With that proof of concept in hand, we then investigated the possibility of combining the peptide/oligourea hybrid approach with the protraction strategy, which consists of functionalizing a peptide with a fatty acid chain to promote its binding to albumin and prolong its in vivo half-life. Semaglutide is a GLP-1 analogue with an Aib in position 2 and a C18 chain linked to K20 through a small PEG spacer which was accepted by the FDA in 2018 for the treatment of diabetes as a once weekly treatment and is presently in phase 3 clinical trial for a once daily oral treatment.^{50,51} We synthesized hybrid **24** which is a semaglutide analogue with an oligourea fragment replacing the four last amino acids (semaglutide-[A^uA^uA^u]²⁸⁻³¹) like in hybrid **22**. In vitro (cAMP production functional assay), hybrid **24** was found to be equally potent to semaglutide (supplementary data, Figure S87). The pharmacodynamics properties of **24** were next assessed in a study on *db/db* mice using FDA approved exenatide, liraglutide and semaglutide as positive controls. A single dose of analogues or placebo was injected intra peritoneal and the blood glucose was followed over time. After 48 h, hybrid **24** and semaglutide were still active while exenatide and liraglutide showed no more significant activity. Although hybrid **24** and semaglutide showed similar activities in vitro and in vivo (Figure 7), it is noteworthy that hybrid **24** displays a longer half-life than semaglutide in mouse plasma (160 vs 11 h, supplementary data, Figure S84) and pancreatin (3.2 vs 0.63 h, Figure 7-C). Because pancreatin degradation assays are relevant to predict the stability of peptides in the gastrointestinal tract,⁵² our data thus suggest that hybrid **24** could be suitable for oral administration and the peptide/oligourea hybrid approach reported here could be used to improve oral delivery of some peptide-based drugs.”*

Figure 7 | Blood glucose study in db/db mice. (A-B) Blood glucose concentration in db/db mice (male, 40-45 g, n = 5) before and after dosing: 200 µg/kg (50 nmol/kg) intra peritoneal. Formulation: 20 µg/mL in PBS 1X. The dosing was done at T0. (B) AUC of the IPGTT curves. (C) Pancreatin degradation assay. (two-way anova and Bonferroni post test: * p<0.05; ** p<0,01; *** p<0,001; one way anova with Dunnett's multiple comparison test: # p<0,05; ### p<0,001)

REVIEWERS' COMMENTS:

Reviewer #1 (Remarks to the Author):

I said that the previous version was fine for Nat. Comm. so of course I think that this version is fine too. The authors show GLP-1 analogs with urea parts that have extended half-life in mice. The reason for the extended half-life is a mystery. Although the authors have pulled back on earlier claims about resistance to proteases, the text still seems to say that this is the reason for extended half-life. To me the protease data say the opposite. But the observations in the paper are interesting, even if no one understands why the peptides last a long time in vivo. It is time to publish the paper in Nat. Comm. so the community can judge it.

Reviewer #3 (Remarks to the Author):

1) Regarding the "concept extension" problem, I still insist on my opinion. Although there are several different aspects just as claimed by authors, the idea of design of α -Peptide-Oligourea Hybrids to mimic α -peptide and the helix structure is not brand new, and it comes from the previous results and findings of " α -Peptide-Oligourea Chimeras" which can be seen in the paper "Angew. Chem., 2015, 127, 9954–9958".

2) Since the authors chose GLP-1 as an example to validate the effectiveness of their approach of peptide/oligourea hybrids, at least I think, it is necessary to compare the obtained GLP analogues with existing drugs based on GLP-1. Of course, from the available results in this manuscript, we could conclude that the GLP-1 analogues described in this manuscript would lose their hypoglycemic effect after 6 hours in vivo.

Combination with the data of semaglutide derivatives added by the authors and the data from GLP analogues, I think that peptide/oligourea hybrids would be an alternative or supplementary approach to the modification of other peptide drugs for improving pharmacodynamic and pharmacokinetic properties. Therefore, here I give a positive evaluation of this manuscript for publishing in Nat. Comm.

3) Authors should make a revision on following expression:

"the peptide/oligourea hybrid approach reported here could be used to improve oral delivery of some peptide-based drugs."

I think this is overstated, it would be appropriate to be expressed as "the peptide/oligourea hybrid approach reported here would be used to improve pharmacokinetic properties of some peptide-based drugs."

Reviewers' Comments (plus author responses):

Reviewer #1 (Remarks to the Author):

I said that the previous version was fine for Nat. Comm. so of course I think that this version is fine too. The authors show GLP-1 analogs with urea parts that have extended half-life in mice. The reason for the extended half-life is a mystery. Although the authors have pulled back on earlier claims about resistance to proteases, the text still seems to say that this is the reason for extended half-life. To me the protease data say the opposite. But the observations in the paper are interesting, even if no one understands why the peptides last a long time in vivo. It is time to publish the paper in Nat. Comm. so the community can judge it.

We are grateful to the reviewer for acknowledging that our work "is stronger than before" We agree that the exact cause of the extended half-life is still open to debate, however we think that better resistance to proteases is playing a role, although it is hard to know to which extend. It is also possible that these hybrids modify the distribution in the organism or the signaling pathways in the cells. Those hypotheses will have to be examined further and will be reported in due course.

We are grateful again to the reviewer for recognizing the value of our work.

Reviewer #3 (Remarks to the Author):

1) Regarding the "concept extension" problem, I still insist on my opinion. Although there are several different aspects just as claimed by authors, the idea of design of α -Peptide-Oligourea Hybrids to mimic a-peptide and the helix structure is not brand new, and it comes from the previous results and findings of " α -Peptide-Oligourea Chimeras" which can be seen in the paper "Angew. Chem., 2015, 127, 9954–9958".

2) Since the authors chose GLP-1 as an example to validate the effectiveness of their approach of peptide/oligourea hybrids, at least I think, it is necessary to compare the obtained GLP analogues with existing drugs based on GLP-1. Of course, from the available results in this manuscript, we could conclude that the GLP-1 analogues described in this manuscript would lose their hypoglycemic effect after 6 hours in vivo.

Combination with the data of semaglutide derivatives added by the authors and the data from GLP analogues, I think that peptide/oligourea hybrids would be an alternative or supplementary approach to the modification of other peptide drugs for improving pharmacodynamic and pharmacokinetic properties. Therefore, here I give a positive evaluation of this manuscript for publishing in Nat. Comm.

We are grateful to the reviewer to 'give a positive evaluation of this manuscript for publishing in Nat. Comm.'

3) Authors should make a revision on following expression:

“the peptide/oligourea hybrid approach reported here could be used to improve oral delivery of some peptide-based drugs.”

I think this is overstated, it would be appropriate to be expressed as “the peptide/oligourea hybrid approach reported here would be used to improve pharmacokinetic properties of some peptide-based drugs.”

The expression was modified accordingly.